# Character-level Convolutional Networks for Text Classification[*]

**Xiang Zhang**      **Junbo Zhao**      **Yann LeCun**
Courant Institute of Mathematical Sciences, New York University
719 Broadway, 12th Floor, New York, NY 10003
{xiang, junbo.zhao, yann}@cs.nyu.edu

## Abstract

This article offers an empirical exploration on the use of character-level convolutional networks (ConvNets) for text classification. We constructed several large-scale datasets to show that character-level convolutional networks could achieve state-of-the-art or competitive results. Comparisons are offered against traditional models such as bag of words, n-grams and their TFIDF variants, and deep learning models such as word-based ConvNets and recurrent neural networks.

## 1   Introduction

Text classification is a classic topic for natural language processing, in which one needs to assign predefined categories to free-text documents. The range of text classification research goes from designing the best features to choosing the best possible machine learning classifiers. To date, almost all techniques of text classification are based on words, in which simple statistics of some ordered word combinations (such as n-grams) usually perform the best [12].

On the other hand, many researchers have found convolutional networks (ConvNets) [17] [18] are useful in extracting information from raw signals, ranging from computer vision applications to speech recognition and others. In particular, time-delay networks used in the early days of deep learning research are essentially convolutional networks that model sequential data [1] [31].

In this article we explore treating text as a kind of raw signal at character level, and applying temporal (one-dimensional) ConvNets to it. For this article we only used a classification task as a way to exemplify ConvNets' ability to understand texts. Historically we know that ConvNets usually require large-scale datasets to work, therefore we also build several of them. An extensive set of comparisons is offered with traditional models and other deep learning models.

Applying convolutional networks to text classification or natural language processing at large was explored in literature. It has been shown that ConvNets can be directly applied to distributed [6] [16] or discrete [13] embedding of words, without any knowledge on the syntactic or semantic structures of a language. These approaches have been proven to be competitive to traditional models.

There are also related works that use character-level features for language processing. These include using character-level n-grams with linear classifiers [15], and incorporating character-level features to ConvNets [28] [29]. In particular, these ConvNet approaches use words as a basis, in which character-level features extracted at word [28] or word n-gram [29] level form a distributed representation. Improvements for part-of-speech tagging and information retrieval were observed.

This article is the first to apply ConvNets only on characters. We show that when trained on large-scale datasets, deep ConvNets do not require the knowledge of words, in addition to the conclusion

---

[*]An early version of this work entitled "Text Understanding from Scratch" was posted in Feb 2015 as arXiv:1502.01710. The present paper has considerably more experimental results and a rewritten introduction.

from previous research that ConvNets do not require the knowledge about the syntactic or semantic structure of a language. This simplification of engineering could be crucial for a single system that can work for different languages, since characters always constitute a necessary construct regardless of whether segmentation into words is possible. Working on only characters also has the advantage that abnormal character combinations such as misspellings and emoticons may be naturally learnt.

## 2   Character-level Convolutional Networks

In this section, we introduce the design of character-level ConvNets for text classification. The design is modular, where the gradients are obtained by back-propagation [27] to perform optimization.

### 2.1   Key Modules

The main component is the temporal convolutional module, which simply computes a 1-D convolution. Suppose we have a discrete input function $g(x) \in [1, l] \rightarrow \mathbb{R}$ and a discrete kernel function $f(x) \in [1, k] \rightarrow \mathbb{R}$. The convolution $h(y) \in [1, \lfloor (l - k + 1)/d \rfloor] \rightarrow \mathbb{R}$ between $f(x)$ and $g(x)$ with stride $d$ is defined as

$$h(y) = \sum_{x=1}^{k} f(x) \cdot g(y \cdot d - x + c),$$

where $c = k - d + 1$ is an offset constant. Just as in traditional convolutional networks in vision, the module is parameterized by a set of such kernel functions $f_{ij}(x)$ ($i = 1, 2, \ldots, m$ and $j = 1, 2, \ldots, n$) which we call *weights*, on a set of inputs $g_i(x)$ and outputs $h_j(y)$. We call each $g_i$ (or $h_j$) input (or output) *features*, and $m$ (or $n$) input (or output) feature size. The outputs $h_j(y)$ is obtained by a sum over $i$ of the convolutions between $g_i(x)$ and $f_{ij}(x)$.

One key module that helped us to train deeper models is temporal max-pooling. It is the 1-D version of the max-pooling module used in computer vision [2]. Given a discrete input function $g(x) \in [1, l] \rightarrow \mathbb{R}$, the max-pooling function $h(y) \in [1, \lfloor (l - k + 1)/d \rfloor] \rightarrow \mathbb{R}$ of $g(x)$ is defined as

$$h(y) = \max_{x=1}^{k} g(y \cdot d - x + c),$$

where $c = k - d + 1$ is an offset constant. This very pooling module enabled us to train ConvNets deeper than 6 layers, where all others fail. The analysis by [3] might shed some light on this.

The non-linearity used in our model is the rectifier or thresholding function $h(x) = \max\{0, x\}$, which makes our convolutional layers similar to rectified linear units (ReLUs) [24]. The algorithm used is stochastic gradient descent (SGD) with a minibatch of size 128, using momentum [26] [30] 0.9 and initial step size 0.01 which is halved every 3 epoches for 10 times. Each epoch takes a fixed number of random training samples uniformly sampled across classes. This number will later be detailed for each dataset sparately. The implementation is done using Torch 7 [4].

### 2.2   Character quantization

Our models accept a sequence of encoded characters as input. The encoding is done by prescribing an alphabet of size $m$ for the input language, and then quantize each character using 1-of-$m$ encoding (or "one-hot" encoding). Then, the sequence of characters is transformed to a sequence of such $m$ sized vectors with fixed length $l_0$. Any character exceeding length $l_0$ is ignored, and any characters that are not in the alphabet including blank characters are quantized as all-zero vectors. The character quantization order is backward so that the latest reading on characters is always placed near the begin of the output, making it easy for fully connected layers to associate weights with the latest reading.

The alphabet used in all of our models consists of 70 characters, including 26 english letters, 10 digits, 33 other characters and the new line character. The non-space characters are:

```
abcdefghijklmnopqrstuvwxyz0123456789
-,;.!?:'''/\|_@#$%^&*~`+-=<>()[]{}
```

Later we also compare with models that use a different alphabet in which we distinguish between upper-case and lower-case letters.

## 2.3 Model Design

We designed 2 ConvNets – one large and one small. They are both 9 layers deep with 6 convolutional layers and 3 fully-connected layers. Figure 1 gives an illustration.

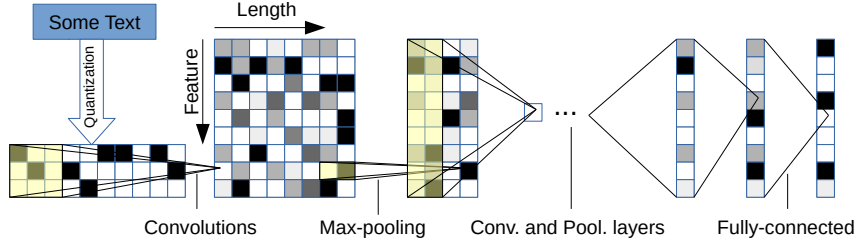

Figure 1: Illustration of our model

The input have number of features equal to 70 due to our character quantization method, and the input feature length is 1014. It seems that 1014 characters could already capture most of the texts of interest. We also insert 2 dropout [10] modules in between the 3 fully-connected layers to regularize. They have dropout probability of 0.5. Table 1 lists the configurations for convolutional layers, and table 2 lists the configurations for fully-connected (linear) layers.

Table 1: Convolutional layers used in our experiments. The convolutional layers have stride 1 and pooling layers are all non-overlapping ones, so we omit the description of their strides.

| Layer | Large Feature | Small Feature | Kernel | Pool |
|---|---|---|---|---|
| 1 | 1024 | 256 | 7 | 3 |
| 2 | 1024 | 256 | 7 | 3 |
| 3 | 1024 | 256 | 3 | N/A |
| 4 | 1024 | 256 | 3 | N/A |
| 5 | 1024 | 256 | 3 | N/A |
| 6 | 1024 | 256 | 3 | 3 |

We initialize the weights using a Gaussian distribution. The mean and standard deviation used for initializing the large model is $(0, 0.02)$ and small model $(0, 0.05)$.

Table 2: Fully-connected layers used in our experiments. The number of output units for the last layer is determined by the problem. For example, for a 10-class classification problem it will be 10.

| Layer | Output Units Large | Output Units Small |
|---|---|---|
| 7 | 2048 | 1024 |
| 8 | 2048 | 1024 |
| 9 | Depends on the problem | |

For different problems the input lengths may be different (for example in our case $l_0 = 1014$), and so are the frame lengths. From our model design, it is easy to know that given input length $l_0$, the output frame length after the last convolutional layer (but before any of the fully-connected layers) is $l_6 = (l_0 - 96)/27$. This number multiplied with the frame size at layer 6 will give the input dimension the first fully-connected layer accepts.

## 2.4 Data Augmentation using Thesaurus

Many researchers have found that appropriate data augmentation techniques are useful for controlling generalization error for deep learning models. These techniques usually work well when we could find appropriate invariance properties that the model should possess. In terms of texts, it is not reasonable to augment the data using signal transformations as done in image or speech recognition, because the exact order of characters may form rigorous syntactic and semantic meaning. Therefore,

the best way to do data augmentation would have been using human rephrases of sentences, but this is unrealistic and expensive due the large volume of samples in our datasets. As a result, the most natural choice in data augmentation for us is to replace words or phrases with their synonyms.

We experimented data augmentation by using an English thesaurus, which is obtained from the `mytheas` component used in LibreOffice[1] project. That thesaurus in turn was obtained from Word-Net [7], where every synonym to a word or phrase is ranked by the semantic closeness to the most frequently seen meaning. To decide on how many words to replace, we extract all replaceable words from the given text and randomly choose $r$ of them to be replaced. The probability of number $r$ is determined by a geometric distribution with parameter $p$ in which $P[r] \sim p^r$. The index $s$ of the synonym chosen given a word is also determined by a another geometric distribution in which $P[s] \sim q^s$. This way, the probability of a synonym chosen becomes smaller when it moves distant from the most frequently seen meaning. We will report the results using this new data augmentation technique with $p = 0.5$ and $q = 0.5$.

## 3   Comparison Models

To offer fair comparisons to competitive models, we conducted a series of experiments with both traditional and deep learning methods. We tried our best to choose models that can provide comparable and competitive results, and the results are reported faithfully without any model selection.

### 3.1   Traditional Methods

We refer to traditional methods as those that using a hand-crafted feature extractor and a linear classifier. The classifier used is a multinomial logistic regression in all these models.

**Bag-of-words and its TFIDF**. For each dataset, the bag-of-words model is constructed by selecting 50,000 most frequent words from the training subset. For the normal bag-of-words, we use the counts of each word as the features. For the TFIDF (term-frequency inverse-document-frequency) [14] version, we use the counts as the term-frequency. The inverse document frequency is the logarithm of the division between total number of samples and number of samples with the word in the training subset. The features are normalized by dividing the largest feature value.

**Bag-of-ngrams and its TFIDF**. The bag-of-ngrams models are constructed by selecting the 500,000 most frequent n-grams (up to 5-grams) from the training subset for each dataset. The feature values are computed the same way as in the bag-of-words model.

**Bag-of-means on word embedding**. We also have an experimental model that uses k-means on word2vec [23] learnt from the training subset of each dataset, and then use these learnt means as representatives of the clustered words. We take into consideration all the words that appeared more than 5 times in the training subset. The dimension of the embedding is 300. The bag-of-means features are computed the same way as in the bag-of-words model. The number of means is 5000.

### 3.2   Deep Learning Methods

Recently deep learning methods have started to be applied to text classification. We choose two simple and representative models for comparison, in which one is word-based ConvNet and the other a simple long-short term memory (LSTM) [11] recurrent neural network model.

**Word-based ConvNets**. Among the large number of recent works on word-based ConvNets for text classification, one of the differences is the choice of using pretrained or end-to-end learned word representations. We offer comparisons with both using the pretrained word2vec [23] embedding [16] and using lookup tables [5]. The embedding size is 300 in both cases, in the same way as our bag-of-means model. To ensure fair comparison, the models for each case are of the same size as our character-level ConvNets, in terms of both the number of layers and each layer's output size. Experiments using a thesaurus for data augmentation are also conducted.

**Long-short term memory**. We also offer a comparison with a recurrent neural network model, namely long-short term memory (LSTM) [11]. The LSTM model used in our case is word-based, using pretrained word2vec embedding of size 300 as in previous models. The model is formed by taking mean of the outputs of all LSTM cells to form a feature vector, and then using multinomial logistic regression on this feature vector. The output dimension is 512. The variant of LSTM we used is the common

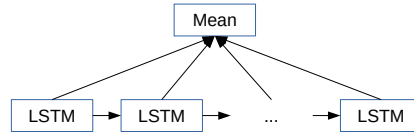

Figure 2: long-short term memory

"vanilla" architecture [8] [9]. We also used gradient clipping [25] in which the gradient norm is limited to 5. Figure 2 gives an illustration.

## 3.3 Choice of Alphabet

For the alphabet of English, one apparent choice is whether to distinguish between upper-case and lower-case letters. We report experiments on this choice and observed that it usually (but not always) gives worse results when such distinction is made. One possible explanation might be that semantics do not change with different letter cases, therefore there is a benefit of regularization.

## 4 Large-scale Datasets and Results

Previous research on ConvNets in different areas has shown that they usually work well with large-scale datasets, especially when the model takes in low-level raw features like characters in our case. However, most open datasets for text classification are quite small, and large-scale datasets are splitted with a significantly smaller training set than testing [21]. Therefore, instead of confusing our community more by using them, we built several large-scale datasets for our experiments, ranging from hundreds of thousands to several millions of samples. Table 3 is a summary.

Table 3: Statistics of our large-scale datasets. Epoch size is the number of minibatches in one epoch

| Dataset | Classes | Train Samples | Test Samples | Epoch Size |
|---------|---------|---------------|--------------|------------|
| AG's News | 4 | 120,000 | 7,600 | 5,000 |
| Sogou News | 5 | 450,000 | 60,000 | 5,000 |
| DBPedia | 14 | 560,000 | 70,000 | 5,000 |
| Yelp Review Polarity | 2 | 560,000 | 38,000 | 5,000 |
| Yelp Review Full | 5 | 650,000 | 50,000 | 5,000 |
| Yahoo! Answers | 10 | 1,400,000 | 60,000 | 10,000 |
| Amazon Review Full | 5 | 3,000,000 | 650,000 | 30,000 |
| Amazon Review Polarity | 2 | 3,600,000 | 400,000 | 30,000 |

**AG's news corpus**. We obtained the AG's corpus of news article on the web[2]. It contains 496,835 categorized news articles from more than 2000 news sources. We choose the 4 largest classes from this corpus to construct our dataset, using only the title and description fields. The number of training samples for each class is 30,000 and testing 1900.

**Sogou news corpus**. This dataset is a combination of the SogouCA and SogouCS news corpora [32], containing in total 2,909,551 news articles in various topic channels. We then labeled each piece of news using its URL, by manually classifying the their domain names. This gives us a large corpus of news articles labeled with their categories. There are a large number categories but most of them contain only few articles. We choose 5 categories – "sports", "finance", "entertainment", "automobile" and "technology". The number of training samples selected for each class is 90,000 and testing 12,000. Although this is a dataset in Chinese, we used `pypinyin` package combined with `jieba` Chinese segmentation system to produce Pinyin – a phonetic romanization of Chinese. The models for English can then be applied to this dataset without change. The fields used are title and content.

Table 4: Testing errors of all the models. Numbers are in percentage. "Lg" stands for "large" and "Sm" stands for "small". "w2v" is an abbreviation for "word2vec", and "Lk" for "lookup table". "Th" stands for thesaurus. ConvNets labeled "Full" are those that distinguish between lower and upper letters

| Model | AG | Sogou | DBP. | Yelp P. | Yelp F. | Yah. A. | Amz. F. | Amz. P. |
|---|---|---|---|---|---|---|---|---|
| BoW | 11.19 | 7.15 | 3.39 | 7.76 | 42.01 | 31.11 | 45.36 | 9.60 |
| BoW TFIDF | 10.36 | 6.55 | 2.63 | 6.34 | 40.14 | 28.96 | 44.74 | 9.00 |
| ngrams | 7.96 | 2.92 | 1.37 | **4.36** | 43.74 | 31.53 | 45.73 | 7.98 |
| ngrams TFIDF | **7.64** | **2.81** | **1.31** | 4.56 | 45.20 | 31.49 | 47.56 | 8.46 |
| Bag-of-means | **16.91** | **10.79** | **9.55** | **12.67** | **47.46** | **39.45** | **55.87** | **18.39** |
| LSTM | 13.94 | 4.82 | 1.45 | 5.26 | 41.83 | 29.16 | 40.57 | 6.10 |
| Lg. w2v Conv. | 9.92 | 4.39 | 1.42 | 4.60 | 40.16 | 31.97 | 44.40 | 5.88 |
| Sm. w2v Conv. | 11.35 | 4.54 | 1.71 | 5.56 | 42.13 | 31.50 | 42.59 | 6.00 |
| Lg. w2v Conv. Th. | 9.91 | - | 1.37 | 4.63 | 39.58 | 31.23 | 43.75 | 5.80 |
| Sm. w2v Conv. Th. | 10.88 | - | 1.53 | 5.36 | 41.09 | 29.86 | 42.50 | 5.63 |
| Lg. Lk. Conv. | 8.55 | 4.95 | 1.72 | 4.89 | 40.52 | 29.06 | 45.95 | 5.84 |
| Sm. Lk. Conv. | 10.87 | 4.93 | 1.85 | 5.54 | 41.41 | 30.02 | 43.66 | 5.85 |
| Lg. Lk. Conv. Th. | 8.93 | - | 1.58 | 5.03 | 40.52 | 28.84 | 42.39 | 5.52 |
| Sm. Lk. Conv. Th. | 9.12 | - | 1.77 | 5.37 | 41.17 | 28.92 | 43.19 | 5.51 |
| Lg. Full Conv. | 9.85 | 8.80 | 1.66 | 5.25 | 38.40 | 29.90 | 40.89 | 5.78 |
| Sm. Full Conv. | 11.59 | 8.95 | 1.89 | 5.67 | 38.82 | 30.01 | 40.88 | 5.78 |
| Lg. Full Conv. Th. | 9.51 | - | 1.55 | 4.88 | 38.04 | 29.58 | 40.54 | 5.51 |
| Sm. Full Conv. Th. | 10.89 | - | 1.69 | 5.42 | **37.95** | 29.90 | 40.53 | 5.66 |
| Lg. Conv. | 12.82 | 4.88 | 1.73 | 5.89 | 39.62 | 29.55 | 41.31 | 5.51 |
| Sm. Conv. | 15.65 | 8.65 | 1.98 | 6.53 | 40.84 | 29.84 | 40.53 | 5.50 |
| Lg. Conv. Th. | 13.39 | - | 1.60 | 5.82 | 39.30 | **28.80** | 40.45 | **4.93** |
| Sm. Conv. Th. | 14.80 | - | 1.85 | 6.49 | 40.16 | 29.84 | **40.43** | 5.67 |

**DBPedia ontology dataset**. DBpedia is a crowd-sourced community effort to extract structured information from Wikipedia [19]. The DBpedia ontology dataset is constructed by picking 14 non-overlapping classes from DBpedia 2014. From each of these 14 ontology classes, we randomly choose 40,000 training samples and 5,000 testing samples. The fields we used for this dataset contain title and abstract of each Wikipedia article.

**Yelp reviews**. The Yelp reviews dataset is obtained from the Yelp Dataset Challenge in 2015. This dataset contains 1,569,264 samples that have review texts. Two classification tasks are constructed from this dataset – one predicting full number of stars the user has given, and the other predicting a polarity label by considering stars 1 and 2 negative, and 3 and 4 positive. The full dataset has 130,000 training samples and 10,000 testing samples in each star, and the polarity dataset has 280,000 training samples and 19,000 test samples in each polarity.

**Yahoo! Answers dataset**. We obtained Yahoo! Answers Comprehensive Questions and Answers version 1.0 dataset through the Yahoo! Webscope program. The corpus contains 4,483,032 questions and their answers. We constructed a topic classification dataset from this corpus using 10 largest main categories. Each class contains 140,000 training samples and 5,000 testing samples. The fields we used include question title, question content and best answer.

**Amazon reviews**. We obtained an Amazon review dataset from the Stanford Network Analysis Project (SNAP), which spans 18 years with 34,686,770 reviews from 6,643,669 users on 2,441,053 products [22]. Similarly to the Yelp review dataset, we also constructed 2 datasets – one full score prediction and another polarity prediction. The full dataset contains 600,000 training samples and 130,000 testing samples in each class, whereas the polarity dataset contains 1,800,000 training samples and 200,000 testing samples in each polarity sentiment. The fields used are review title and review content.

Table 4 lists all the testing errors we obtained from these datasets for all the applicable models. Note that since we do not have a Chinese thesaurus, the Sogou News dataset does not have any results using thesaurus augmentation. We labeled the best result in blue and worse result in red.

# 5  Discussion

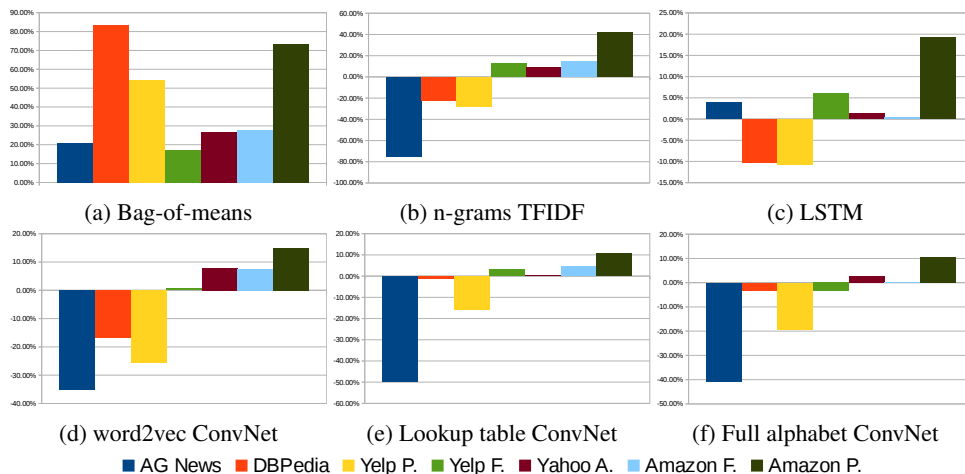

(a) Bag-of-means     (b) n-grams TFIDF     (c) LSTM

(d) word2vec ConvNet     (e) Lookup table ConvNet     (f) Full alphabet ConvNet

■ AG News   ■ DBPedia   ■ Yelp P.   ■ Yelp F.   ■ Yahoo A.   ■ Amazon F.   ■ Amazon P.

Figure 3: Relative errors with comparison models

To understand the results in table 4 further, we offer some empirical analysis in this section. To facilitate our analysis, we present the relative errors in figure 3 with respect to comparison models. Each of these plots is computed by taking the difference between errors on comparison model and our character-level ConvNet model, then divided by the comparison model error. All ConvNets in the figure are the large models with thesaurus augmentation respectively.

**Character-level ConvNet is an effective method**. The most important conclusion from our experiments is that character-level ConvNets could work for text classification without the need for words. This is a strong indication that language could also be thought of as a signal no different from any other kind. Figure 4 shows 12 random first-layer patches learnt by one of our character-level ConvNets for DBPedia dataset.

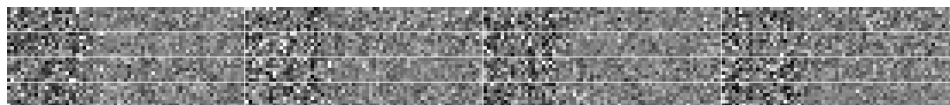

Figure 4: First layer weights. For each patch, height is the kernel size and width the alphabet size

**Dataset size forms a dichotomy between traditional and ConvNets models**. The most obvious trend coming from all the plots in figure 3 is that the larger datasets tend to perform better. Traditional methods like n-grams TFIDF remain strong candidates for dataset of size up to several hundreds of thousands, and only until the dataset goes to the scale of several millions do we observe that character-level ConvNets start to do better.

**ConvNets may work well for user-generated data**. User-generated data vary in the degree of how well the texts are curated. For example, in our million scale datasets, Amazon reviews tend to be raw user-inputs, whereas users might be extra careful in their writings on Yahoo! Answers. Plots comparing word-based deep models (figures 3c, 3d and 3e) show that character-level ConvNets work better for less curated user-generated texts. This property suggests that ConvNets may have better applicability to real-world scenarios. However, further analysis is needed to validate the hypothesis that ConvNets are truly good at identifying exotic character combinations such as misspellings and emoticons, as our experiments alone do not show any explicit evidence.

**Choice of alphabet makes a difference**. Figure 3f shows that changing the alphabet by distinguishing between uppercase and lowercase letters could make a difference. For million-scale datasets, it seems that not making such distinction usually works better. One possible explanation is that there is a regularization effect, but this is to be validated.

**Semantics of tasks may not matter**. Our datasets consist of two kinds of tasks: sentiment analysis (Yelp and Amazon reviews) and topic classification (all others). This dichotomy in task semantics does not seem to play a role in deciding which method is better.

**Bag-of-means is a misuse of word2vec** [20]. One of the most obvious facts one could observe from table 4 and figure 3a is that the bag-of-means model performs worse in every case. Comparing with traditional models, this suggests such a simple use of a distributed word representation may not give us an advantage to text classification. However, our experiments does not speak for any other language processing tasks or use of word2vec in any other way.

**There is no free lunch**. Our experiments once again verifies that there is not a single machine learning model that can work for all kinds of datasets. The factors discussed in this section could all play a role in deciding which method is the best for some specific application.

# 6 Conclusion and Outlook

This article offers an empirical study on character-level convolutional networks for text classification. We compared with a large number of traditional and deep learning models using several large-scale datasets. On one hand, analysis shows that character-level ConvNet is an effective method. On the other hand, how well our model performs in comparisons depends on many factors, such as dataset size, whether the texts are curated and choice of alphabet.

In the future, we hope to apply character-level ConvNets for a broader range of language processing tasks especially when structured outputs are needed.

# Acknowledgement

We gratefully acknowledge the support of NVIDIA Corporation with the donation of 2 Tesla K40 GPUs used for this research. We gratefully acknowledge the support of Amazon.com Inc for an AWS in Education Research grant used for this research.

## Footnotes

[1]`http://www.libreoffice.org/`

[2]http://www.di.unipi.it/~gulli/AG_corpus_of_news_articles.html

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
