[Reviews · NeurIPS 2015]

Submitted by Assigned_Reviewer_1

I have read the author response to reviewer comments.
Summary: The paper is an empirical analysis of character level CNN's for text classification, which has already been investigated in NLP and speech conferences.

Submitted by Assigned_Reviewer_2

line 034: is essentially convolutional network -> is essentially a convolutional network line 036: temploral -> temporal line 039: several these datasets -> several of these dataset line 041: has be -> has been line 130: texts of interests -> texts of interest line 146: We initialization -> We initialize line 236: such distinguish -> such distinction line 243: in the text classification -> for text classification line 245: hundreds thousands -> hundreds of thousands line 260: 4 largest classes -> the 4 largest classes line 262: for each classes -> for each class line 262: and testing 1900 -> and testing is 1900 line 358: learnt one -> learnt by one line 382: such distinguish -> such distinction
Summary: The paper presents a convolutional neural network architecture for text classification which works directly at the character level (different from the usual methods which work with words). They present a comprehensive experimental comparison of their approach (with many different settings) to other approaches for text classification (both traditional bag-of-word approaches and using word embeddings). For the experiments they use large datasets ranging from hundreds of thousands of samples to millions of samples, which the authors have curated themselves, so the reported results they compare against are their own (and not from published papers). The results are very encouraging and they are able to show that character-level CNNs are very competitive for large datasets (most of them in English and one in Chinese). However, it would have been nice if they also used some benchmarks from literature instead of comparing only against their own experiments with other methods. Also, the paper is poorly written in terms of English grammar. I have listed below some of the mistakes they made.

Submitted by Assigned_Reviewer_3

This paper deals with a character-based inference model for text classification. The model is a temporal 1-D convolutional neural network using a "one-hot" character representation and a fully connected decision layer on top of it. The model is empirically compared with 2 other popular models: LSTM and word-based CNN. Furthermore, the empirical contribution of unsupervised pretraining approach word2vec is investigated. Another claimed contribution is the production of 8 large annotated corpora extracted from news feeds. The paper pretty well-written but the scope of the contribution is not clear. The contribution seems mainly empirical. Indeed, a question would be the contribution of this work compared to [1] that is not mentioned in the references. Indeed, character based temporal CNN model and thesaurus enrichment have been proposed in this previous work. In fact, the authors seems to not mention clearly the potential theoretical contribution of their work, presenting it as a "empirical exploration". Another remark could be that text classification remains the most basic task of Statistical NLP, what about structured prediction like parsing or even sequence labeling task where dataset do exist ?

refs : [1] Text Understanding from Scratch, Xiang Zhang, Yann LeCun, 2015.
Summary: The paper is well-written but the definition of the actual contribution is not clear. Indeed, the contribution seems mainly experimental and empirical, the question would be the contribution of this work compared to [1] that is not mentioned in the references.

refs : [1] Text Understanding from Scratch, Xiang Zhang, Yann LeCun, 2015.

Submitted by Assigned_Reviewer_4

The paper presents a character-level convolutional network architecture and applies it to eight text classification problems on large datasets that the authors construct. It also presents comparative results from several word-based deep NN models as well as bag-of-ngrams models. The character-level ConvNets outperform word-based models on four out of eight datasets, when word-based data augmentation is used.

The clarity and quality of writing are ok but the presentation of the method and results could have been much more clear. There are numerous grammatical and spelling errors.

The originality is not very high since character-level models have been used for other tasks. The representation of the input as a fixed length vector of length 1014 characters is unintuitive, as one would like to have a story for handling documents of all lengths.

The use of data augmentation using a thesaurus is not consistent with the stated advantage of character-level models: if words and even a thesaurus is available, then word-level tokenization can be performed.

The experimental results contain a large number of comparison models as well as several variants of the proposed architecture: large vs small hidden layers, use of thesaurus augmentation, and lower-case characters versus both lower and upper-case ones.

For four of the eight problems character-level models with data augmentation performed best and outperformed word-based models. In some cases character-level models without data augmentation outperformed word-level models as well. The data settings where character-level models performed best were also ones with user-generated content which might contain misspellings and ones with the largest number of training instances.

Since the hyper-parameters of the different models were not tuned and there is no comparison to published state-of-the-art results (or models with state-of-the-art results), it is not clear what the significance of these experiments could be.

*****

Thanks for the detailed response. It will be good to add the justification for dataset choice to the paper.

Summary: The paper presents an experimental study of character-level ConvNet models for text classification in comparison with word-based models on eight datasets. On the largest and noisiest datasets, character-level models with thesaurus-based data augmentation outperform word-level models. The significance of the work is unclear as there are no comparisons with state-of-the-art models that use both word and character-level information.

Submitted by Assigned_Reviewer_5

This paper gives experimental studies of character-level convolutional networks. The empirical results are compared to word-level models and other classical text classification models such as bag-of-words, n-grams and so on. This paper only reports the experimental comparison results, there is limited novelty though it is well-written and easy to follow.

The strong points are:

(1) Comprehensive experiments: 22 models (including variants of models) were tested on 8 datasets.

(2) An important finding that character-level ConvNet model has comparable results to word-level models.

The weak points are: (1) In Section 5 (Discussion), most claims are only observations but not conclusions, they are in lack of validation. The only results given are the error rates, some of claims are too arbitrary, e.g. Bag-of-means is a misuse of word2vec. I am not opposing this claim but it definitely needs more evidence to make such a strong claim. Maybe the bad results of Bag-of-means are because of the influence of K or other parameters such as the dimension of word vectors.

(2) Sogou corpus is a corpus of Chinese news. The model does not use the Chinese language directly but use its Romanized sounding system (pinyin) of Chinese. The review would suggest to use the Chinese characters directly instead of the pinyin.

(3) Figure 3 are not clear in printed version.

Summary: This paper gives empirical comparison studies of character-level ConvNets to other classical text classification models. The conclusion is that the character-level ConvNets have comparable results.

Author Feedback
Author rebuttal: We want to express our gratitude for the hardworking of the reviewers and chairs. The reviews are of high quality and represent community perception of our paper. They contain many detailed suggestions that we could use to make the article better. We also hope to take this opportunity to clarify some possible misunderstandings.

Concerning comparisons with previous state-of-the art papers, given the fact that ConvNets require large-scale datasets to work well, we found that there are very few suitable datasets freely available for doing so. Also, in many previously published results, large scale datasets are splitted in an imbalance way such that is impossible to compare. One example is RCV1 [1], the LYRL2004 split only uses roughly 20,000 samples for training and all the rest nearly 780,000 are used for testing. Therefore, instead of confusing our community even more by re-using these datasets with a different split, we built several new datasets from scratch and compared many representative methods for them.

We strive to not over-claim for our work. Therefore we are explicit that this is an empirical study. In section 5 we point out explicitly when something is to be verified by further study. As for the case of bag-of-centroids as a misuse of word2vec, we later discovered a paper [2] that offers much complete study to prove our point. Their study shows that word2vec works well when we add some structures rather than just naive k-means to the model. We will add it to our references.

A complete tuning of hyper-parameters was not feasible computationally for this research. That said, we did our best in choosing models that are fair for comparisons. These efforts include ensuring sufficiently large feature sizes (up to 500,000) for linear classification models, using consistent model capacity between word-based and character-based models, and extending experiments to ConvNets models of at least 2 different sizes.

We agree with the reviewers' suggestion that using a thesaurus is not perfectly consistent with the advantage of character-level models. However, it does not indicate that word-based method is better. Augmentation in noisy dataset (in terms of character combinations) could still be useful because many words could still be correct.

We also acknowledge ther reviewers' opinion that using a Romanization of Chinese is not a canonical approach in the context of our models. This is one of the things that we are working on. The difficulty is that one-of-n encoding is infeasible because there are too many Chinese characters. We hope to complete a follow-up work on this with a solution.

Text classification is undoubted the simplest task in NLP. We hope for more work in this direction tackling more complicated NLP tasks.

Finally, there is one other misunderstanding regarding contribution. We feel that is is more appropriate to communicate this to the area/program chairs to protect the anonymity of our double-blind review system.

Again, thanks a lot for the reviewers' detailed comments! They are very helpful for us.

References:
[1] David D. Lewis, Yiming Yang, Tony G. Rose, Fan Li, RCV1: A New Benchmark Collection for Text Categorization Research, Journal of Machine Learning Research 5 (2004) 361-397
[2] G. Lev, B. Klein, L. Wolf. In Defense of Word Embedding for Generic Text Representation. NLDB, 2015.